# The Effect of Rare Earths on the Response of Photo UV-Activate ZnO Gas Sensors

**DOI:** 10.3390/s22218150

**Published:** 2022-10-25

**Authors:** Isabel Sayago, José Pedro Santos, Carlos Sánchez-Vicente

**Affiliations:** Institute for Physical and Information Technologies (ITEFI-CSIC), 28006 Madrid, Spain

**Keywords:** nanoparticles ZnO, rare earth (Er; Tb; Eu; Dy), resistive gas sensors, pollutant gases (NO_2_, CO and CH_4_), UV-light irradiation, room temperature

## Abstract

In this work, ZnO nanoparticle resistive sensors decorated with rare earths (REs; including Er, Tb, Eu and Dy) were used at room temperature to detect atmospheric pollutant gases (NO_2_, CO and CH_4_). Sensitive films were prepared by drop casting from aqueous solutions of ZnO nanoparticles (NPs) and trivalent RE ions. The sensors were continuously illuminated by ultraviolet light during the detection processes. The effect of photoactivation of the sensitive films was studied, as well as the influence of humidity on the response of the sensors to polluting gases. Comparative studies on the detection properties of the sensors showed how the presence of REs increased the response to the gases detected. Low concentrations of pollutant gases (50 ppb NO_2_, 1 ppm CO and 3 ppm CH_4_) were detected at room temperature. The detection mechanisms were then discussed in terms of the possible oxidation-reduction (redox) reaction in both dry and humid air atmospheres.

## 1. Introduction

Currently, air pollution monitoring is carried out using air quality monitoring stations that provide accurate data; however, these stations are scarce, since their location and administration require considerable investment [1]. The standard reference measurement equipment of the stations consists of optical and chemical analyzers [1]. They are complex and bulky, with high acquisition and maintenance costs [2,3]. Each instrument costs between five thousand and tens of thousands of euros. The cost of calibration and maintenance must also be added [3]. Air quality monitoring stations are often located in limited areas of the cities, which makes it difficult to collect representative and reliable information from the whole urban area. Therefore, they do not provide global information that can help to properly manage potential pollution problems [4].

Gas sensors are the best alternative to acquire more information about air quality in order to obtain global contamination maps that allow the proper management of pollution. In particular, resistive type gas sensors based on semiconductor metal oxides (MOX), such as ZnO, SnO_2_ and TiO_2_, with low costs, easy production, a compact size and simple measurement electronics are the most widely used [5,6,7,8]. The detection process is characterized by resistance changes in the sensitive layer (semiconductor material) as a function of the surrounding atmosphere. However, MOX-resistive sensors typically operate at high temperatures, which limits their application as sensitive materials and leads to material instability, increased power consumption and response drifts [9,10]. 

UV light activation is an effective way to improve the performance of sensors, as it can facilitate the development of low-power and low-cost portable devices [11]. The first research into photoactivation of sensors was carried out at the end of the last century [12,13,14], but it has only gained prominence in the last decades. This has been made possible by the development of new, smaller and low-cost light-emitting diode (LED) devices incorporated into test cells, allowing sensor devices to operate at room temperature [11,15,16].

ZnO is one of the most used semiconductors as a sensitive layer in gas sensors due to its good response, low cost, good thermal and chemical stability and easy production. ZnO is an n-type semiconductor with a bandgap of 3.37 eV, and a large excitation binding energy of 60 meV [17]. It also exhibits a high conductivity and optical transparency in UV-VIS, and can be decorated with rare earth elements to improve its optical and electromechanical properties, acquiring great potential as a sensitive layer in gas devices [18,19]. 

Rare earth (RE) elements have been used as semiconducting oxide dopants in numerous applications due to their unique optical, magnetic and catalytic properties [20,21,22,23,24]. The basic reason for these peculiar characteristics lies in their electronic configuration, and specifically, in their capacity to generate a wide variety of 4f → 4f type electronic transitions [20,25]. 

Recently, Hastir et al. reported that doping ZnO with RE (Tb^3+^, Dy^3+^ and Er^3+^) increased both the surface basicity and oxygen vacancy concentration, causing lattice distortion [19,26]. Furthermore, the dopants can help to reduce particle size and change its morphology from microrods to nanoparticles [27].

The illumination of zinc oxide with UV light, of which photon energy is equal to or higher than 3.37 eV (ZnO bandgap), enables electrons in the valence band to be rapidly excited to the conduction band, generating a large number of photoinduced electron-hole pairs. These photoelectrons generated by UV light can react to oxygen molecules in the air and promote the formation of photoinduced oxygen ions [16]. The maximum wavelength (λ_max_) that can be used for activation is 368 nm, and it is calculated according to the following equation [17].
λ_max_ ≤ *hc*/E_g_ ≈ 368 nm(1)
where *h* is Planck’s constant (4.14 × 10^−15^ eVs), *c* is the velocity of light (3.00 × 10^8^ ms^−1^) and E_g_ is the forbidden band of ZnO (3.37 eV).

In addition, nanostructured metal oxides with small sizes, confined dimensions and adaptive architectures are the most promising materials for gas sensing due to their special characteristics of high surface-to-volume ratio, high surface reactivity and unique electrical properties [28,29,30]. However, the fabrication processes involve considerable additional costs, and extremely difficult and expensive processes for developing devices based on nanostructure materials [6,30]. Drop-casting is a simple, inexpensive and versatile preparation technique that allows the deposition of multiple materials on relatively small surfaces as the active zone of microsensors [31,32]. The process is conditioned by the surface tension, the volatility of the solvent used, the wettability of the surface to be deposited on, the composition of the solution, the drop impact velocity and the drop size [33]. This method is still widely used in the fabrication of sensitive wearable gas sensor films [33,34,35].

In this study, we present the response of different ZnO-NPs gas sensors, some of them decorated with REs (Tb Er, Dy and Eu), which have been prepared by a simple process, such as drop-casting. Detections were carried out at ambient temperature under UV lighting in both dry and humid air atmospheres. The response of the sensors to pollutant gases, such as NO_2_, CO and CH_4_, are studied. Compared to ZnO, the sensors decorated with REs presented improvements in the sensor benefits with lower resistances and a higher response. The effects of lighting-UV and humidity in the response of the sensors are considered. The detection mechanisms are discussed according to the possible redox reactions that could take place in detection conditions. 

## 2. Materials and Methods

### 2.1. Marerials

Nanoparticles zinc oxide (<100 nm particles size), Erbium (III) chloride hexahydrate, Europium (III) chloride hexahydrate, Terbium (III) chloride, Dysprosium (III) chloride hexahydrate, Europium (III) chloride hexahydrate and Deionized (DI) water were purchased from Sigma Aldrich (Madrid, Spain), and used without any further purification.

### 2.2. Sensors Preparation

The sensitive layers were obtained by drop-casting nanoparticle dispersions on polymeric substrates (FR-4). The FR-4 substrates were manufactured by the company Eurocircuits NV (Mechelen, Belgium). FR-4 is a glass-reinforced epoxy laminate material, commonly used for printed circuit boards (PCBs) because of its flame resistance (self-extinguishing), near zero water absorption and wide operating temperature range (from 50 °C to 115 °C) [36]. 

The FR-4 substrates used were circular with a diameter and thickness of 15 mm and 1.55 mm, respectively. Their surface had four interdigitated electrodes (IDTs), each IDT contained four finger pairs with a finger width and spacing of 0.23 mm and 0.1 mm, respectively. Therefore, four sensitive layers (four sensors) could be deposited and electrically characterized on a FR-4 substrate; the active area of each sensor being 2.87 × 2.87 mm^2^. 

An automatic “dropcaster” device was used for the preparation of the different sensitive films. The device had been developed by our research group and allows the preparation of sensitive layers on the small surface of the sensor. The device had been designed to control the process parameters, such as droplet size, deposition time, waiting time between each droplet and deposition volume. The device characteristics and operating conditions are described in the previous work [37]. All samples were prepared in the same conditions. The dispersion quantity employed was 14 μL, and it was added to the substrate drop by drop. Each drop volume was 200 nL, and the waiting time between drops was 120 s. Throughout the process the sample was illuminated by an LED array which allows the heating of the sample surface between 65–70 °C.

To prepare the sensitive layer, ZnO-NPs were dispersed by sonication in deionized water and then deposited by drop-casting. During deposition, the substrate was illuminated by an infrared LED array (ILR-09B09 UA) to facilitate the solvent evaporation. The different process parameters (droplet size, amount of deposited, time, etc.) were optimized to ensure a homogeneous distribution of the nanoparticles on the substrate. After testing different solvents (water, ethanol, isopropanol and acetone) and conditions, water was chosen as the most suitable solvent, and the dispersion concentration was set at 0.25 mg L^−1^. The RE-decorated samples were obtained by incorporating a certain amount of the precursor (5% wt, rare earth chloride) into the ZnO-NPs dispersions. Table 1 shows the tested sensors and the characteristics of the sensing films. 

### 2.3. Gas Detection Setup

In order to evaluate the response of the sensors to different atmospheric pollutants (NO_2_, CH_4_, CO), the sensors were electrically characterized in a controlled atmosphere (Figure 1). The sensors were placed in a small stainless steel test cell (0.85 cm^3^) with a gas inlet/outlet. The UV-LED (OCU–400 UB355, λ = 353–360 nm, OSA Opto Light GmbH, Berlin, Germany) was mounted above the sensors so that all the light fell on them in a controlled and uniform way, with reduced power consumption. 

Detections were performed at room temperature both in dry and humid air (50% relative humidity-RH), with a constant gas flow of 100 mL·min^−1^. The exposure time to the target gas was 15 min, while the recovery time was 30 min. The reference gases used were air (99.999% purity), CH_4_ (10 ppm, diluted in air), CO (10 ppm, diluted in air) and NO_2_ (1 ppm, diluted in air), purchased from Nippon Gases España, S.L.U.

A gas mixing unit (GMU, Ray IE, Cáceres, Spain) with mass flow meters was used to control the gas mixture concentration. Mass flow controllers allow for the modification of the flow rate of the reference gases to obtain the target gas concentrations. A humidity generator was also available to mix dry and humid air and, controlling the flow of both, the desired humidity was achieved. In addition, in the pre-chamber, where the mixing of the different gases took place, there was a humidity and temperature sensor that took continuous measurements and allowed the relative humidity of the gas mixture to be established.

Sensor responses, given by the change of their electrical resistance, were measured with an electrometer (6517 model, Keithley, Cleveland, OH, USA). The processes and equipment were controlled through a PC, using a home-developed software with LabVIEW that monitors and displays the resistance of the sensors as a function of time and the surrounding atmosphere.

## 3. Results and Discussions

### 3.1. Material Characterization

The sensitive material was characterized by scanning electron microscopy (SEM) and energy-dispersive spectroscopy (EDX) using a Quanta 3D FEG (FEI Company) equipped with an energy dispersive detector. Samples were prepared under the same conditions as the sensors, but on silicon substrates.

SEM was used to examine the surface morphology of the samples. The SEM images of the ZnO-NPs and some RE-decorated samples at different magnifications are shown in Figure 2. It is observed that most of the NPs are quasi-spherical, although there are some rectangular, radial and triangular NPs. In general, dimensions of NPs are less than 100 nm, and only a few of them reach 200 or 300 nm. The NPs exhibit agglomerations, which may be due to their large surface area; the NPs have an affinity for each other, and tend to form randomly distributed asymmetric clusters. However, it is possibly the drop-casting preparation process that leads to the superposition of the NPs, and the formation of arrays of NPs. When REs were added to the starting material, low magnification images showed an additional porous and discontinuous surface layer that resembled sea waves (Figure 2a,b). This is more evident in the Tb-containing samples. At higher magnifications, it can be seen that the morphology of the ZnO-NPs is not significantly affected by the presence of RE, although there appears to be some larger and more elongated particles.

EDX analyses determined the elemental composition of the sample. The EDX spectra indicated the presence of Zn and O as major elements, and confirmed the presence of the REs (Figure 3). The EDX mapping revealed that both the majority elements (Zn and O) and the REs were uniformly distributed on the sample surface. The EDX qualitative analyses presented results similar to those found in the literature for ZnO nanoparticles [38], and confirmed the atomic ratio Zn/O was close to 1.

### 3.2. Electrical Characterization

First, the effect of UV-LED illumination on the resistance of the sensors was considered. For this purpose, alternate on/off cycles of the LED were carried out in both a dry and humid (50% RH) air atmosphere. In both atmospheres, resistance changes were similar. The illumination with UV light produced a drastic decrease in the resistance of all sensors (four orders of magnitude), and quickly returned to the initial resistance value when the sensors were no longer illuminated. These resistance variations due to ultraviolet light have been reported in highly resistive samples [39].

The resistance variations of the sensitive material can be attributed to the loss of both physisorbed and chemisorbed oxygen from the surface of the nanoparticles. The NPs have a high surface area, which favors the adsorption of oxygen species on their surface. In addition, due to their small size, there are multiple active centers at the grain boundaries of the NPs for oxygen adsorption. This leads to very high initial resistance values for the sensors in the air. When the UV-LED light is on, oxygen photodesorption takes place. Subsequently, in the off cycle (no illumination), the oxygen will be adsorbed again and the sensors will recover their initial resistance value. It should be noted that oxygen desorption during photoactivation is very fast and it requires illumination from 2 to 3 min for the resistance of the sensors to reach a stable value, while the recovery time in the dark is longer. As already mentioned, illumination with light generates electron-hole pairs, and rapid changes in charge carriers. Whereas in the dark, charge carrier recombination is slower due to the high number of active sites of the material that prevent charge carrier recombination [40].

In general, RE-decorated sensors have initially higher resistance values than non-decorated ZnO sensors. Only Dy-decorated samples present a similar resistance to ZnO samples. The RE decoration of sensors induces a conductance decrease due to reduction of charge carriers. REs, being electron-rich, provide more active places for oxygen adsorption, producing a sensor resistance increase. Figure 4 shows how, without illumination, all sensors decorated with RE tend to have values considerably higher than ZnO sensors. The exception is Dy, with a resistance value similar to non-decorated sensors. This can be explained in terms of RE basicity. This RE basicity does not gradually decrease with the increase of the atomic number, but it decreases according to the following sequence: Eu > Tb~Er > Dy [41]. Changes in resistance follow the trend of the RE basicity: the lower the basicity, the lower the resistance change. Under UV-LED light irradiation, the resistance of the sensors decreases by four orders of magnitude in the RE-Eu,Er,Tb-decorated samples, and two orders of magnitude in the non-decorated and Dy-decorated ones (Figure 4). Photoactivation enables the sensors to achieve stable resistance values in the range of the measuring equipment. The incorporation of REs seems to favor the photoactivation of the ZnO, as evidenced by the variations in the resistance reached during illumination in the RE-decorated sensors. It is known that rare earth metals can trap photoinduced electrons, thereby reducing the recombination of electron-hole pairs and increasing the photocatalytic activity [25,42].

The performance of the sensors during the detection processes was evaluated in terms of sensor response. In this work, “Response” is defined by the relative resistance change, as follows:Reducing gases   Response (%) = (R_a_ − R_g_/R_g_) × 100
Oxidizing gases   Response (%) = (R_g_ − R_a_/R_a_) × 100
where R_a_ is the initial resistance of the sensor in air atmosphere and R_g_ is the resistance measured after being exposed to test gas.

The feasibility of the sensors was determined by high responses and fast detection processes. In this sense, the response and recovery times were the main parameters that determined the detection capability of the sensor. In the present study, we considered the sensor response time as the time required for the sensor to reach 90% of the maximum variation of the resistance when exposed to the target gas. The time required for the sensor to recover 90% of its original resistance was considered the sensor recovery time.

The detection efficacy of the sensors was evaluated at room temperature in air atmosphere under different conditions (dry and humid air; with and without UV-LED illumination).

Initially, detections were carried out in the air without illuminating the sensors. In these conditions, the sensors exhibited high resistances with unstable values. In the dark, the sensors only detected NO_2_ and the detection processes were irreversible; i.e., they did not recover the initial resistance value, as can be seen in Figure 5. NO_2_ detection processes are associated with slow kinetic chemical interactions, which can mean long recovery times or irreversible processes, as observed in the tested sensors. Due to the large amount of interaction energy in chemisorptions, MOX-resistive sensors require an activation energy. This energy is usually a thermal one and allows for high and fast responses. Accordingly, this type of sensor generally works/operates at high temperatures (above 100 °C). The detection mechanisms section will describe the type of interactions between NO_2_ and the sensor surface by means of the redox reactions involved in the detection process.

Subsequently, detections were carried out under a UV illumination of the sensors. The illumination activated the detection processes and the sensors exhibited responses to all the gases tested, detecting concentrations as low as 0.1 ppm for NO_2_, 3 ppm for CO and 5 ppm for CH_4_ (Figure 6 and Figure 7).

#### 3.2.1. Detection in Dry Air

In a dry air atmosphere and under UV illumination, the sensors were exposed to different tested gases in a concentration range from 0.1 to 1 ppm for NO_2_, and 1 to 10 ppm for CO and CH_4_ (Figure 6). The experimental detection curves (the resistance variation with respect to time in the different detection atmospheres) show how the sensor responses gradually increased with the gas concentration (Figure 6a–c). The detection processes were reversible (the resistance initial value was recovered), and the response and recovery times were just a few minutes. 

In general, as was predictable for a n-type (ZnO) semiconductor, its resistance increased with the exposure to NO_2_ oxidizing gas (Figure 6a) and decreased in the presence of reducing gases, such as CO and CH_4_ (Figure 6b,c, respectively). As a remarkable observation, abnormal responses were obtained (resistance slightly increased) when the sensor was exposed to CH_4_ concentrations lower than or equal to 5 ppm (Figure 6c,d).

In all cases, the introduction of REs improved the response of ZnO sensors to the tested gases (Figure 6). As can be seen in Figure 5, all sensors showed remarkably higher responses to NO_2_ than to CO and CH_4_, in spite of their significant higher concentrations. 

The best response to NO_2_ was reached by the ZnO-Dy sensor and, according to the RE- functionalization of the sensors, the best responses obtained were: Dy > Er > Eu > Tb. In the CO case, the response variations of the ZnO sensors due to the presence of REs were not so remarkable, and the best responses were reached by sensors decorated with Er and Tb (similar values), followed by the sensors with Dy and Eu. 

Investigations carried out by Hastir et al. [19] regarding ethanol detection by ZnO sensors doped with RE showed how the sensor response was related to the RE basicity: the lower the RE basicity, the higher the sensor response. The response of our sensors was also associated with the RE basicity. The results proved that Dy (lower basicity)-decorated sensors showed better responses to the different tested gases.

As previously noted, abnormal behavior has been observed in CH_4_ detection. General sensors present low responses to this gas, and only the sensors decorated with Dy and Tb show a resistance decrease in the presence of 5 ppm (probably the limit concentration detected by the sensors) through to 7 ppm onwards; every sensor shows a resistance decrease, as can be observed in Figure 6c. 

#### 3.2.2. Detection in Humid Air

The effect of humidity on the sensitive properties of sensors was only considered under illumination because, in the dark, the sensors were not viable due to their poor sensitivity and lack of reversibility in detection processes.

Under UV illumination, the experiments were carried out in an air atmosphere with a relative humidity of 50%. The respective highest concentrations of the sensors were exposed to: 0.5 ppm for NO_2_, 5 ppm for CO and 5 ppm for CH_4_; i.e., half the concentration of reference gases (bottles of reference gases in dry air).

The sensors showed good stability during the electric characterization processes, and detections of the target gases were carried out for approximately two months. Unfortunately, the ZnO-Dy sensor, after several detections in humid air, presented an unstable behavior and its NO_2_ detections could not be quantified. 

The experimental detection curves (Figure 7a–c) showed how the responses of the sensors were, in all cases, better than those obtained in dry air. The detection processes were still reversible and the resistance modifications depended on the concentration of the gas detected. They detected concentrations lower than those in dry air. According to the sensors, they detected 0.05 ppm of NO_2_, 1 ppm of CO and 3 ppm of CH_4_. 

Comparing the responses of the sensors in the presence of 50% RH (Figure 7d), it was observed that the sensor decorated with Tb achieved the respective best responses to all the gases tested; however, in NO_2_ detection, the response was of the same order of magnitude as for the sensors decorated with Eu. In CO detection, similar responses were obtained for sensors decorated with Tb, Er and Dy.

Comparing the responses of sensors in both atmospheres, dry and humid air (Figure 7a,b), it was observed that the presence of humidity significantly increased the response to NO_2_ and CH_4_. However, in CO detection, a neglectable increase was observed; in Eu-decorated sensors, a slight decrease in the response was observed.

The calibration curves were also obtained from the plot of sensor responses (%) versus gas concentration. The calibration curves for all sensors fitted second degree polynomial functions with regression coefficients, in most cases higher than 0.95 (R^2^). As an example, the curves corresponding to sensor ZnO-Tb (5% wt) obtained in both atmospheres are shown (Figure 8a–c). 

The presence of humidity did not induce modifications to the sensor response and recovery times, as can be seen in the curves presented in Figure 7 and Figure 8. For reducing gases, the response time was less than 1 min for CH_4_, and from 2 to 3 min for CO. Both presented very fast recovery times, less than 30 s. On the other hand, NO_2_ presented recovery times higher than response ones, with values corresponding to 10 min and 5 min, respectively. 

The low NO_2_ concentrations detected (50 ppb) were remarkable, but the concentrations corresponding to reducing gases (1 ppm of CO and 3 ppm of CH_4_) were noticeable. As far as we know, there are no works concerning resistive sensors detecting at ambient temperature concentrations lower than these (CO and CH_4_). Regarding NO_2_, there have been several published papers about resistive-type MOX sensors for detecting this gas, as can be seen in selected reviews [43,44,45,46]. However, concentrations usually detected are higher than 5 ppm and, in just a few, concentrations lower than 1 ppm are detected. Accordingly, NiO [47] and WO_3_ [48] sensors are able to detect 372 ppb and 160 ppb, respectively, with both type of sensors illuminated by visible light. Although they detected low concentrations, of the responses obtained, all case results have usually been lower than those presented in this work. Recent work based on heterojunctions with 2D materials, such as graphene or MXene, seem to show results more in line with our sensors, though their main limitations pertain to their complex and long preparation methods [49,50,51,52]. A summary of the detection parameters corresponding to some low temperature sensors for NO_2_ detection is shown in Table 2.

It is known that water vapor behaves as a reducing gas and produces a sensor conductivity increase in the air atmosphere [53,54]. Our experimental data supported this behavior, and our sensors presented a slight conductivity increase in humid air. 

In humid air, a competence among molecules in the environment (O_2_, H_2_O) is established. In general, H_2_O molecules are adsorbed on the sensor surface active sites. They can block the adsorption of O_2_ molecules, reducing the sensor response [54,55,56]. Recent research has proved how UV illumination can improve H_2_O desorption, minimizing or reducing its effect. This has been shown in works by Hyodo et al., using SnO_2_ sensors [57], and by Wang and et al., using ZnO sensors decorated with In_2_0_3_ [58]. However, the behavior observed in our sensors under illumination (with a significant increase in their response to gases in humid air) seems to be more powerful because water possibly behaves as an oxygen source to replace the lost oxygen [59]. This would justify the improvement in the response observed in our sensors in the presence of moisture.

Photocatalytic peculiarities and redox of RE are important to promote surface properties and electron transfer. The RE presence improves the response of the sensors to detected gases due to the surface area increase, the forbidden band energy decrease and ultimately, to the highest adsorption capacity [60]. The EDX analyses showed how rare earths were uniformly distributed on the sensing film surface (Figure 3). This uniform distribution allows an increase of the available active sites to adsorb the gases, thereby increasing the response of the sensors [60,61].

Lastly, it should be noted that sensor stability and repeatability studies have been carried out; as shown in the Appendix A. The sensors had good stability, and their detection curves presented similar paths. The detection processes were repetitive and the resistance variations (due to sensor responses to gases) were always similar (same order of magnitude) for every concentration of a certain gas (see the Appendix A).

#### 3.2.3. Selectivity of Sensors

The selective detection of specific gases is still a challenge for the commercial application of a gas sensor based on metallic oxides. To investigate selectivity, we carried out detection with three pollutant gases, one oxidizing (NO_2_) and two reducing (CO, CH_4_). Sensor selectivity was defined as the ratio of the maximum response of the interference gas to the maximum response of the target gas.
(% Selectivity) = (Response_other gas_/Response_target gas_) × 100%

In this work, the tested sensors presented a high response to NO_2_ (Figure 8a). The responses in the humid environment of the ZnO-Eu and ZnO-Tb sensors at NO_2_ 0.3 ppm were ~55 and ~128 times higher, respectively, than the response to CH_4_ (5 ppm), and 15 and 69 times higher than the response to CO (5 ppm). Figure 9 shows selectivity to NO_2_ of the different tested sensors with respect to every secondary gas considered (CO and CH_4_). Sensors presented excellent selectivity to NO_2_ with low-cross responses to CH_4_ (R_CH_4__/R_NO_2__ < 5) and CO (R_CO_/R_NO_2__ values of 30 and 5 in dry and humid air, respectively). 

### 3.3. Sensing Mechanism

The most accepted theory, with respect to sensing mechanisms activated by light, considers that these mechanisms are similar to those that take place in the dark, and are due to the oxygen species previously adsorbed from the air atmosphere. The adsorption takes place mainly on the semiconductor surface active points that, in the case of n type semiconductors, are closely related to the semiconductor oxygen vacancies [62,63,64]. In the dark, at temperatures lower than 150 °C, the oxygen is adsorbed as O_2_^−^ [60]. However, at ambient temperature (25 °C), the oxygen molecules have a little possibility of being ionosorbed, though some studies have confirmed the presence of ionized oxygen molecules (O_2_^−^_ads_) on the semiconductor surface [54,65,66]. According to the response of our sensors, without illumination, characterized by slow and irreversible detection, processes seemed to indicate and confirm that the most probable interaction type must be chemisorption, and therefore, it was necessary to identify an energy supply to activate the adsorption/desorption reactions. In contrast, the detection processes under illumination with fast response and recovery times (lower than a minute in some cases) suggested that the interaction mechanism could be due to Van der Waals forces (a weak interaction) between the species, presented on the surface sensor and the detected gases.

Illuminating the sensors under UV light implies two main steps: (1) Photon adsorption with an energy higher than the forbidden band. (2) Generation, separation, migration or recombination of photogenerated hole pairs/electrons.

Through UV illumination, the photoinduced electrons diffuse to the interior of the semiconductor while the photoinduced holes migrate to the surface and react with the chemical adsorbed oxygen species (O_2_^−^_ads_), producing oxygen desorption on the surface, and thus, the sensor resistance decreases as was determined when the resistance modifications of the sensors due to illumination were justified [67,68,69]. Simultaneously, the photoinduced electrons interactuate with the oxygen molecules in the air and generate photoinduced oxygen species (O_2_^−^_(*hυ*)_). According to bibliographic reports, these species are loosely bound to the sensor surface. The possible redox reactions that take place in air atmosphere by UV illumination of the sensors are [16,17,69]:*hυ* → e^−^_(*hυ)*_ + *h*^+^_(*hυ*)_(2)
O_2_^−^_(ads)_ + *h*^+^_(*hυ*)_ → O_2_(3)
O_2_ + e^−^_(*hυ*)_ → O_2_^−^_(*hυ*)_(4)

It is known that light-generated gas molecules are, generally, weakly bound to the material’s surface [48,62,70]. Accordingly, adsorption/desorption of the different species generated by UV-illumination can take place, easily improving the gas detection rate. We considered that the mechanism that prevails and conditions the response of our sensors to detected gases was due to the presence of photoactivated oxygen species (O_2_^−^_(*hυ*)_).

For reducing gases, oxidation takes place and produces CO_2_. The redox reactions that can take place on the surface of the sensor are [71,72]:For CO   O_2_^−^_(*hυ*)_ + CO → CO_2_ + e^−^(5)
For CH_4_   2O_2_^−^
_(*hv*)_ + CH_4_ → CO_2_ + 2H_2_O + 2 e^−^(6)
O_2_^−^_(*hv*)_ + CH_4_ → CH_2_O + H_2_O + e^−^(7)

In the case of CH_4_, a total combustion producing CO_2_ (Equation (5)) or a partial combustion producing formaldehyde (Equation (7)) can take place. In both cases, it is confirmed that the presence of the methyl radicals are an intermediate species [73,74].

In the case of NO_2_ (oxidizing gas), as in the dark, it directly interacts with the semiconductor surface by direct adsorption (Equation (8)); furthermore, due to its electronic affinity (2.27 eV) being higher than its O_2_ affinity (0.44 eV) [75], it can extract electrons from the adsorbed O_2_ species (Equation (9)). In both cases, the electron capture (possibly photogenerated) takes place according to the following redox reactions [43,72]:NO_2_ + e^−^ → NO_2_^−^_(ads)_(8)
NO_2_ + O_2_^−^_(*hv*)_ + 2e^−^ → NO_2_^−^_(ads)_ + 2O^−^
(9)

Other possible reactions involving the formation of nitrates (NO_3_^−^) as intermediate species both in the dark [76] and under UV illumination [77,78]. Under UV illumination, the redox reactions would be [76,78]:NO_2_ + O_2_^−^_(*hv*)_ + e^−^ → NO_3_^−^_(ads)_ + O^−^
(10)

NO_2_ detection research carried out on-site by Wang et al., both in the dark and under UV illumination, proved in both cases the presence of NO_2_^−^ and NO_3_^−^ species on ZnO sensors. However, the proportions of both species were modified according to sensor illumination. The IR spectra obtained showed how the UV illumination led to an NO_2_^−^ species decrease and, on the other hand, to a significant NO_3_^−^ species increase [65]. Therefore, NO_2_ can be adsorbed in different ways (Equations (8)–(10)). Our sensors presented high responses to NO_2_ because the previously described reactions (Equations (8)–(10)) simultaneously took place. They led to a decrease in conductivity in sensing films, and an increase in resistance.

#### Effect of Humidity

Environmental humidity is an important factor that influences the efficiency of sensors, modifying their conductance and responses. The water molecules can be adsorbed by physical adsorption (molecular water) and chemical adsorption (hydroxyl groups) [71].

A mechanism that determines the effect of humidity on the gas response has not yet been established. However, the most accepted hypothesis establishes that water displaces the chemically adsorbed oxygen, reducing the response of sensors. In humid air, it takes place initially, the dissociation of water molecules being ionadsorbed on the sensor surface [55].
HO_2_ → OH^−^ + H^+^(11)

If the sensor is photoactivated, it can determine the neutralization of hydroxyl groups by the photoinduced holes generated, according to [79,80]:OH^−^_(ads)_ + *h*^+^_(*hυ*)_ → OH*(12)

Furthermore, electron paramagnetic resonance (EPR) spectroscopy measurements carried out by Nasriddinov et al. on the surface of SNO_2_ and SNO_2_/Ru confirmed that O_2_^−^ species can be easily replaced by OH* radicals in a humid atmosphere [81]. 

Although humidity is one of the main factors affecting sensor performance, there are few reports detailing the interactions between gases and possible redox reactions. 

The OH* radical is a powerful oxidant that could interact with carbon-containing gases, such as CH_4_ or CO, according to the following reaction [79,80], releasing electrons and decreasing the sensor resistance.
3OH*_(ads)_ + CH_4_ → CO_2_ + H_2_O + 5H^+^ + 5e^−^(13)
OH*_(ads)_ + CO → CO_2_ + H^+^ + e^−^(14)

The previous reactions would justify the response increase to reducing gases observed in our sensors in the presence of humidity. In particular, for CH_4_, the reaction 13 implies a bigger supply of electrons (a bigger conductivity), and therefore, a lower resistance. Accordingly, resistance variations are higher in the presence of CH_4_.

In the case of NO_2_, the interactions are diverse and complex. The hydroxyl groups are the predominant active sites for the adsorption of NO_2_ molecules, but direct NO_2_ adsorption can also occur (Equation (8)). It is believed that the formation of hydrogen bonds between the hydroxyl group and the NO_2_ molecule can contribute to the enhancement of the response to NO_2_ [82]. 

The reactions of NO_2_ with hydroxyl groups are complex, and the overall proposed reaction involves the formation of nitrate, nitric oxide and water, according to reaction [83,84].
3NO_2_ + 2OH^−^ → 2NO_3_^−^ + NO + H_2_O(15)

Interaction with the hydroxyl radical could also occur, according to the reaction [85].
NO_2_ + OH* → NO_3_^−^ + H^+^(16)

The presence of nitrates on the semiconductor surface leads to a decrease in the sensor electrical conductivity, due to the following reaction [86]:NO_3_^−^_(ads)_ + 2H^+^ + e^−^ → NO_2_ + H_2_O(17)

As the NO_2_ molecules are adsorbed on the surface and the previously described reactions take place, electrons are extracted and the sensor resistance increases.

## 4. Conclusions

This study proposed a simple, inexpensive, versatile and potentially scalable method for preparing low-cost sensors, using the drop–casting technique. Our automated drop-casting equipment allowed us to control and reproduce the preparation of the sensitive layer in small localized areas on the sensor device. Through this process, we prepared ZnO sensors and ZnO sensors decorated with RE (Tb, Er, Dy and Eu), and investigated their sensitive properties to pollutant gases.

The integration of UV-LED in the test cell enabled the sensitive layers to be photo-activated, allowing the sensors to operate at room temperature. 

In general, all sensors exhibited a high sensitivity and selectivity to NO_2_, detecting concentrations as low as 50 ppb. Therefore, the response of the ZnO-Tb sensor in humid air, at 300 ppb of NO_2_, was 128 times higher than the response to CH_4_ (5 ppm) and 69 times higher than the response to CO (5 ppm). Additionally, a low-cross response was obtained by interfering gases CO and CH_4_ with a ratio R_CH_4__/R_NO_2__ close to zero for all sensors, and a ratio R_CO_/R_NO_2__ less than 50 for practically all sensors.

Although the sensors were less sensitive to CO and CH_4_, their capability to detect such low concentrations as CO 1 ppm and CH_4_ 3 ppm at room temperature was remarkable.

Humidity has a positive effect on the response of sensors to all tested gases. This is possibly due to the accumulation of hydroxyl groups and hydroxyl radicals that act as adsorption sites/or effective reagent sites to adsorb the gas molecules on the semiconductor surface. Hydroxyl radicals are generated by UV illumination, and enhance the response of the sensors. Sensors such as the ZnO-Tb-decorated sensor reached responses of 154.7% and 54% at NO_2_ 300 and 100 ppb in the presence of 50% RH, while in dry air, the responses were 20% and 60% respectively. For CH_4_, the response also increased in the presence of humidity; therefore, in the 5 ppm detection, the response changed from 1% to 50% for the ZnO-Tb and ZnO-Dy sensors. However, for CO there were no considerable changes in any of the tested sensors.

The mechanism that conditioned the response of our sensors to tested gases was due to the presence of photoactivated species of oxygen and hydroxyl (O_2_ and OH). 

Rare earths increased the response to the tested gases, due to their catalytic activity, basicity, surface area increase and adsorption capacity. The response of the sensors was related to the basicity of rare earths. The results proved that Dy (lower basicity)-decorated sensors showed better responses to the different tested gases. The highest response to NO_2_ was obtained with the Dy, Tb- and Eu-decorated sensors. For reducing gases, the Dy- and Tb-decorated sensors achieved the best results for CH_4_ detection, while the Dy-, Tb- and Er-decorated sensors provided the best responses for CO.

## Figures and Tables

**Figure 1 sensors-22-08150-f001:**
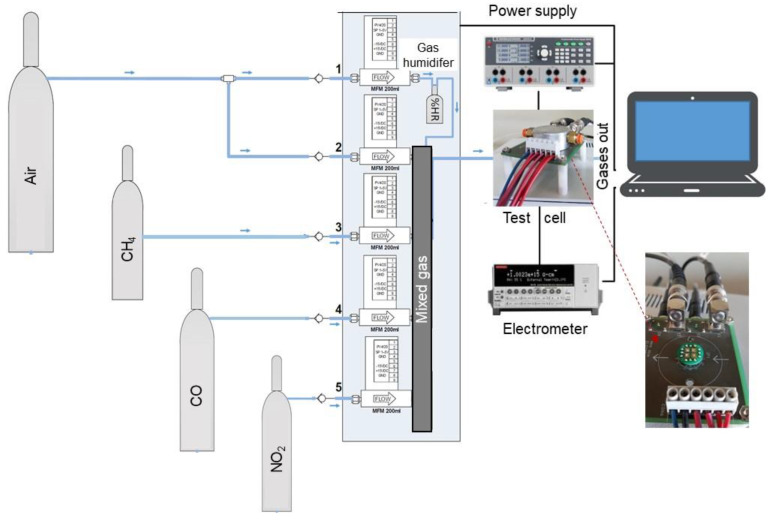
Measurement setup used to measure different gases in real time.

**Figure 2 sensors-22-08150-f002:**
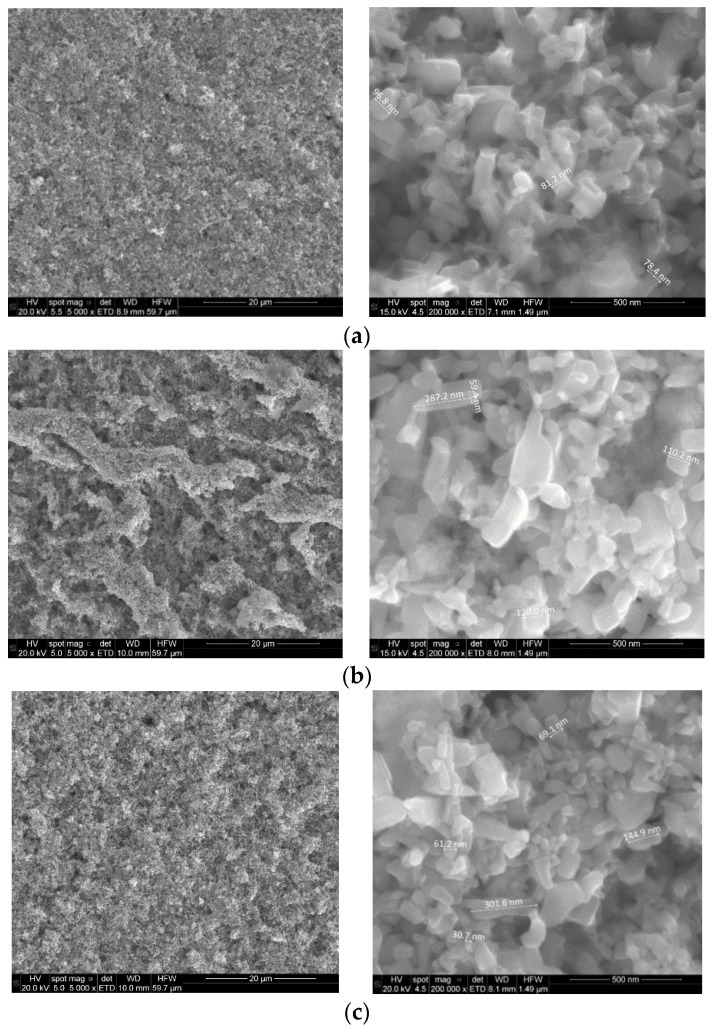
SEM micrographs at different magnifications of some sensitive layers: (**a**) ZnO, (**b**) ZnO-Tb (5%), (**c**) ZnO-Er (5%).

**Figure 3 sensors-22-08150-f003:**
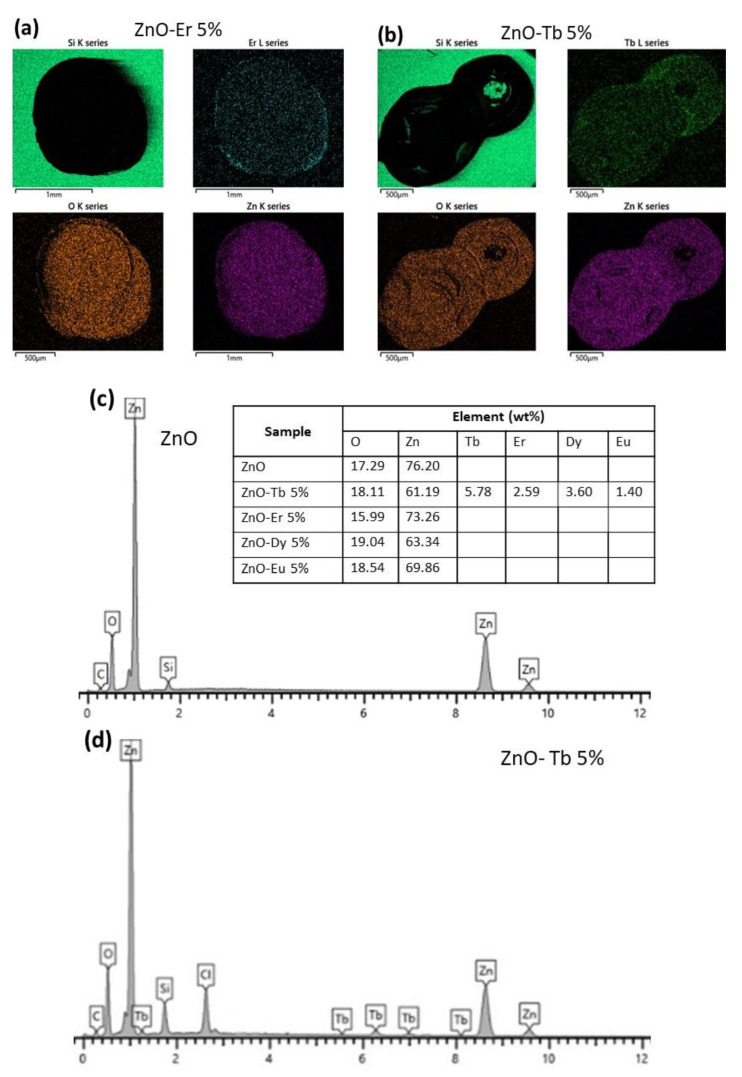
EDX elemental mapping images of some sensitive layers: (**a**) ZnO-Er (5%), (**b**) ZnO-Tb (5%). EDX analysis showing the elemental composition of the samples (table). EDX spectrum of (**c**) ZnO and (**d**) ZnO-Tb (5%).

**Figure 4 sensors-22-08150-f004:**
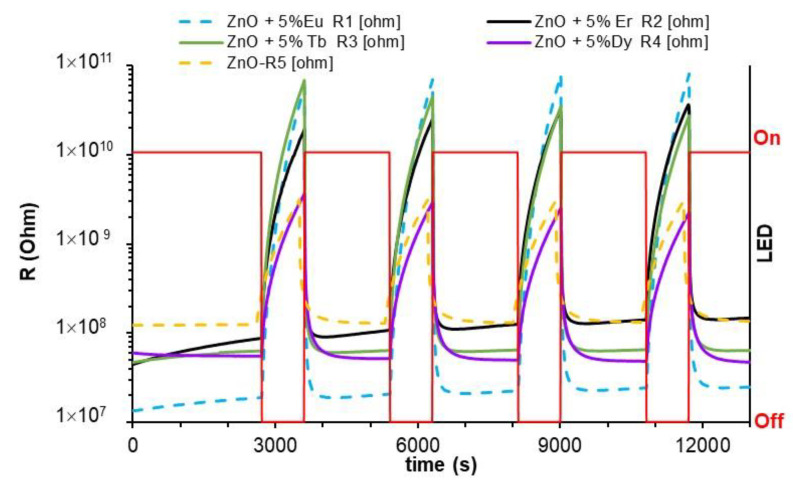
Resistance changes in the different sensors tested with and without UV-LED illumination at room temperature and air atmosphere.

**Figure 5 sensors-22-08150-f005:**
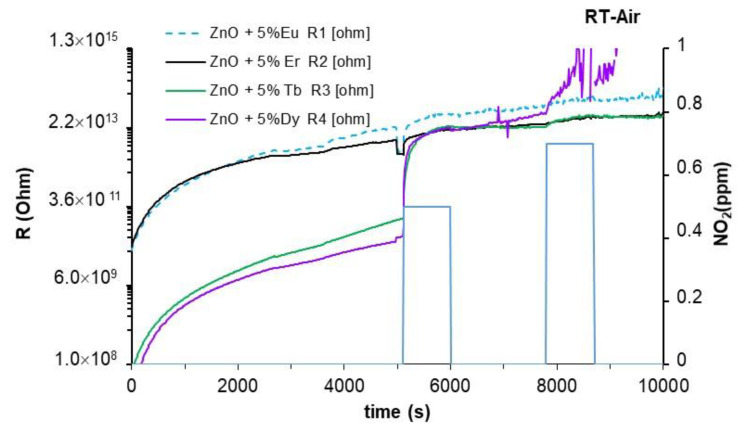
Dynamic response of sensors to different NO_2_ concentrations at room temperature in dry air under darkness condition.

**Figure 6 sensors-22-08150-f006:**
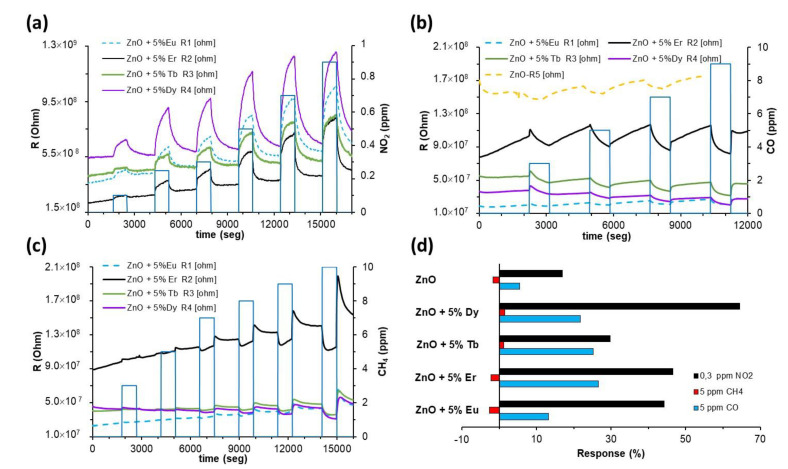
Dynamic response curves of the sensors to pollutant gases: (**a**) NO_2_, (**b**) CO, (**c**) CH_4_ at room temperature in dry air and under UV-LED lighting conditions. (**d**) Responses of the sensors to gases detected in dry air with UV-LED illumination.

**Figure 7 sensors-22-08150-f007:**
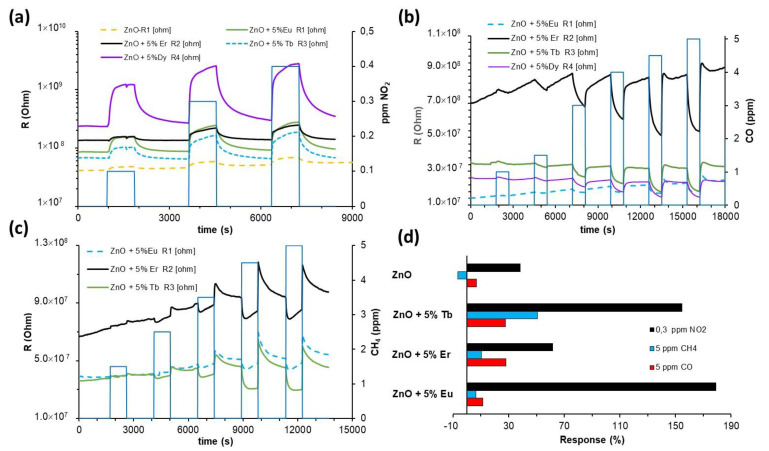
Dynamic response curves of the sensors to pollutant gases: (**a**) NO_2_, (**b**) CO, (**c**) CH_4_ at room temperature in humid air (50% RH) and under UV-LED lighting conditions. (**d**) Responses of the sensors to gases detected in humid air (50% RH) with UV-LED illumination.

**Figure 8 sensors-22-08150-f008:**
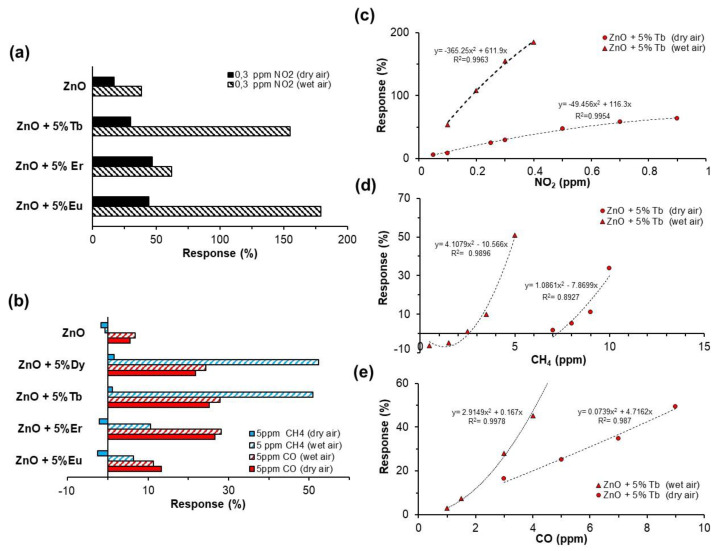
Sensor responses to (**a**) NO_2_ 0.3 ppm, (**b**) CH_4_ 5 ppm and CO 5 ppm, in dry and humid air (50% RH). Calibration curves of ZnO- Tb sensor in dry and humid air (50% RH) to tested gases (**c**) NO_2_, (**d**) CH_4_ and (**e**) CO.

**Figure 9 sensors-22-08150-f009:**
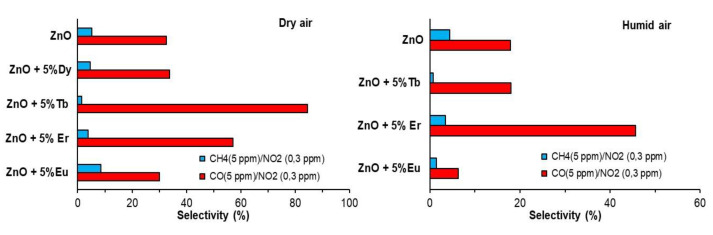
Selectivity for NO_2_ for tested sensors.

**Table 1 sensors-22-08150-t001:** Sensors tested.

Sensors	Sensitive Layer
R1	ZnO (NPs)-Dy (5% wt)
R2	ZnO (NPs)-Eu (5% wt)
R3	ZnO (NPs)-Er (5% wt)
R4	ZnO (NPs)-Tb (5% wt)
R5	ZnO (NPs)

**Table 2 sensors-22-08150-t002:** Sensing properties of the NO_2_ sensors.

Material	Preparation	Response (S%) *	NO_2_ Concetration	Working Temperature	Response/Recovery Time	Reference
ZnO/SnO_2_	Wet chemical method	13.47 ^a^	500 ppb	RT	13.47 min/8 min	[43]
SnO-SnO_2_	hydrothermal method	2.5 ^b^	0.2 ppm	RT	57 s/5 min	[43]
NiO	hydrothermal method	24.2 ^b^	372 ppb	RT (UV)	14.1 min/31.7 min	[47]
WO_3_	sputtering	Not mentioned	160 ppb	RT (illuminated)	Not mentioned	[48]
ZnO-RGO	thermal reduction/soft solution	3.5 ^b^	5 ppm	RT	25 s/15 s	[43]
PtNP-SWCNT	sputtering/thermal treatment	63 ^a^	2 ppm	RT	>180 s/Not mentioned	[44]
SnO_2_-rGO	hydrothermal treatment	3.31 ^b^	3 ppm	50 °C	135 s/200 s	[44]
rGO/ZnO	hydrothermal treatment	119 ^a^	1 ppm	RT	75 s/132 s	[45]
MoS_2_	CVD	23 ^a^	10 ppm	RT	Not mentioned	[45]
WO_3_/S-RGO	one-pot polyol method and (MOD)	50 ^b^	1 ppm	RT (UV)	5 s/56 s	[46]
SnO_-r_GO-OVs	hydrothermal method	4 ^b^	1 ppm	RT	14 s/19 s	[46]
ZnO-NP/Tb	drop-casting	54 ^a^	100 ppb	RT (UV)	2 min/6 min	This work

* Note, ^a^: S (%) = (R_a_ − R_g_)/R_a_ × 100; ^b^: S = (R_g_/R_a_) × 100.

## Data Availability

Not applicable.

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
