# Peer review of "The Effect of Rare Earths on the Response of Photo UV-Activate ZnO Gas Sensors"

_sensors, 2022, doi:10.3390/s22218150_

Round 1

Reviewer 1 Report

Dear Editor

This study investigated the fabrication of resistance-based gas sensor (NO2, CO and CH4 ) using ZnO nanoparticles

1.     Introduction needs new references from Sensors journal

2.     A schematic of the gas sensor and measurement setup needs to be added to the manuscript.

3.     Please add TEM image of ZnO nanoparticles to the manuscript

4.     Equations need references

5.     Initially, detections were carried out in air without illuminating the sensors. In these conditions, the sensors exhibit high resistances with unstable values. In the dark, the sensors only detect NO2 and the detection processes are irreversible, i.e., they do not recover the initial resistance value, as can be seen in Figure 4. The discussion section needs to be improved…more explain details in Fig.4

6.     Physically more explain the behavior of Fig.5(d), why the response of ZnO+5% Dy maximum? Moreover, Fig. 6 (d) too.

7.     Compare the results of this work with other reports in a table

8.     Conclusion should be supported with more data results

Response time and recovery should be defined in the manuscript such as the following references… Performance of gas nanosensor in 1-4 per cent hydrogen concentration, Influence of oxide film surface morphology and thickness on the properties of gas sensitive nanostructure sensor

Author Response

We really thank the reviewer for the comments and suggestions.

Response to reviewer:

  • Introduction needs new references from Sensors journal

Journal SENSORS new references were added

  • A schematic of the gas sensor and measurement setup needs to be added to the manuscript.

A detailed scheme of experimental setup including the different components was incorporated.

  • Please add TEM image of ZnO nanoparticles to the manuscript

Regarding the TEM images of the samples, unfortunately our research group does not have TEM. The characterization by TEM, SEM is carried out by external services requiring an execution time higher than a month. Accordingly, it can not be incorporated to the document in so brief interval.

  • Equations need references

References were added to equations.

  • Initially, detections were carried out in air without illuminating the sensors. In these conditions, the sensors exhibit high resistances with unstable values. In the dark, the sensors only detect NO2and the detection processes are irreversible, i.e., they do not recover the initial resistance value, as can be seen in Figure 4. The discussion section needs to be improved…more explain details in Fig.4.

NO2 detection processes are associated to slow kinetic chemical interactions what it can mean long recovery times or irreversible processes as observed in the tested sensors. Due to the large amount of interaction energy in chemisorptions, MOX resistive sensors require an activation energy. This energy is usually a thermal one and allows to reach high and fast responses. Accordingly, this type of sensors generally works/operates at high temperature (above 100°C). The detection mechanisms section will describe the type of interactions between NO2 and the sensor surface by means of the redox reactions involved in the detection process (see lines 324, page 11). 

  • Physically more explain the behavior of Fig.5(d), why the response of ZnO+5% Dy maximum? Moreover, Fig. 6 (d) too.

Investigations carried out by A. Hastir et al. (19) about ethanol detection by ZnO sensors doped with RE showed how the sensor response was related to the RE basicity: the lower RE basicity the higher sensor response. The response of our sensors is also associated to the RE basicity. The response of our sensors is also associated to the RE basicity. The results proved that Dy (lower basicity) decorated sensors showed better responses to the different tested gases (see lines 324, page 11).

  • Compare the results of this work with other reports in a table

The table was incorporated to the text (Table 2).

  • Conclusion should be supported with more data results. Response time and recovery should be defined in the manuscript such as the following references…Performance of gas nanosensor in 1-4 per cent hydrogen concentration, Influence of oxide film surface morphology and thickness on the properties of gas sensitive nanostructure sensor

Definitions of response and recovery times were included in the document (lines 265-268, page 9)

Conclusions were supported by experimental data     

Reviewer 2 Report

sensors-1985882

First of all, I would like to suggest the authors review the text several times before submitting the article. Redundant sentences and confusion of reference numbers can be seen in your text.

In this manuscript, the authors present the gas response of different ZnO samples decorated with rare earth that are prepared by drop-casting. The authors claim that compared to bare ZnO, the decorated ZnO with REs presented improvements in the gas sensing parameters.

Although the paper has useful information, it needs more data, results, and a closer look. The following comments may help the manuscript to improve:

1-    The order of the references is not correct. For example, after reference [5], there is reference [8].

2-    Remove the last paragraph of the introduction section.

3-    The authors have described the gas sensing setup, but their description is not clear. I recommend providing a schematic of the setup and sensor. This article can be a good example:  DOI: 10.3390/ma15041383.

4-    In the description of figure 1, the authors mentioned, "The size and shape of the nanoparticles change slightly due to the presence of RE." Is this claim just intuitive or has it been measured accurately?

5-    Some of the words in Figure 2 are illegible.

6-    Authors should explain why different RE-decorations lead to different conductance changes.

7-    What is the time unit (seg) in your curves?

8-    The authors have defined the response of the sensor by dividing the two resistance values of the sample. What does the value of the negative response shown in figure 5(d) mean?

9-    Please improve the English grammar and writing skills for this paper.

10- Do you have the same response curve for each test under a certain temperature and humidity?  It is important to have the same response and recovery shape each time you tested for under different conditions. This meant that there was no serious sensing poisoning to your samples and they were fully recoverable to the initial stage. You can put them in the supplementary data.

11- When people talk about the sensor, we care about 3Ss. Sensitivity, Stability and Selectivity.  Do you have any discussion about the selectivity?

12-  The morphology of samples shown in Fig. 1 seems to be in the form of particles. How can the author further obtain such arrays of NPs? Additionally, the cross-sectional morphology of the samples should be given.

Author Response

We really thank the reviewer for the comments and suggestions, which we have implemented in the latest version.

  • The order of the references is not correct. For example, after reference [5], there is reference [8].

The references 6 and 7 have been considered and implemented.

  • Remove the last paragraph of the introduction section.

The paragraph has been removed.

  • The authors have described the gas sensing setup, but their description is not clear. I recommend providing a schematic of the setup and sensor. This article can be a good example:  DOI: 10.3390/ma15041383.

A detailed scheme of experimental setup including the different components was incorporated (Figure 1).

  • In the description of figure 1, the authors mentioned, "The size and shape of the nanoparticles change slightly due to the presence of RE." Is this claim just intuitive or has it been measured accurately?

This claim is an intuitive one that was not possible to corroborate with accurate measuring. Accordingly, it has been removed.

  • Some of the words in Figure 2 are illegible.

Figure 2 (now, Figure 3) has been conveniently amplified.

  • Authors should explain why different RE-decorations lead to different conductance changes.

It was incorporated to the text (lines 223-236) the following explanation about changes in the sensor conductance due to RE presence. The RE decoration of sensors induces a conductance decrease due to reduction of charge carriers. REs, electron rich, provide more active places for oxygen adsorption producing a sensor resistance increase. Figure 4 shows how without illumination all sensors decorated with RE tend to values considerably higher than ZnO sensors. The exception is Dy with a resistance value similar to non-decorated sensors. This can be explained in terms of RE basicity. This RE basicity does not gradually decrease with the increase of the atomic number but it decreases according to the following sequence: Eu>Tb~Er>Dy [44]. Changes in resistance follow the trend of the RE basicity: the lower the basicity the lower resistance change.

  • What is the time unit (seg) in your curves?

In all graphics "seg" was modified to "s" (seconds)

  • The authors have defined the response of the sensor by dividing the two resistance values of the sample. What does the value of the negative response shown in figure 5(d) mean?

There has been a mistake defining the response of the sensors. The definition has been corrected. The response of the sensors presented in this work is the following one:

Reducing gases                Response (%)= (Ra-Rg/Rg)×100

Oxidizing gases                 Response (%)= (Rg-Ra/Ra) ×100

Regarding Figure 5d (now Figure 6d) the represented negative value corresponds to abnormal behaviours observed in the response of the sensors detecting low CH4 (reducing gas) concentrations. The resistance slightly increased instead of decreasing.

  • Please improve the English grammar and writing skills for this paper.

The document has been carefully reviewed. Anyway, if it were accepted the MPDI English editing service would be contracted.

  • Do you have the same response curve for each test under a certain temperature and humidity?  It is important to have the same response and recovery shape each time you tested for under different conditions. This meant that there was no serious sensing poisoning to your samples and they were fully recoverable to the initial stage. You can put them in the supplementary data.

The "Supplementary data" section incorporates some information related to the sensor response in the short and long term.

  • When people talk about the sensor, we care about 3Ss. Sensitivity, Stability and Selectivity.  Do you have any discussion about the selectivity?

A new section has been added to discuss the selectivity of the sensors (see section 3.2.3, page 16).

  • The morphology of samples shown in Fig. 1 seems to be in the form of particles. How can the author further obtain such arrays of NPs? Additionally, the cross-sectional morphology of the samples should be given.

The sensing films have been prepared by drop-casting from aqueous dispersions of ZnO nanoparticles (size of nanoparticles < 100 nm). The process is automatically carried out by the drop-casting equipment designed by the research group. All samples have been prepared in the same conditions. The dispersion quantity employed was 14 μL and it is added to the substrate drop by drop. Each drop volume is 200 nL and the waiting time between drops is 120 s. All along the process the sample is illuminated by a LED array (ILR–09B09 UA) which allows heating the sample surface between 65 - 70 °C. (lines 119-123)

The preparation conditions have been added (lines 119-123, page 3) to the "Preparation section" and Figure 2 (previously, figure 1) includes the proper comments (lines 185-191, page 5).

The NPs exhibit agglomerations, which may be due to their large surface area, the NPs have an affinity for each other and tend to form randomly distributed asymmetric clusters. But it is possibly the drop-casting preparation process what leads to both the superposition of the NPs and the formation of arrays of NPS.

Regarding the cross-sectional morphology of the samples, unfortunately our research group does not have the techniques for sample characterization and analysis. The analysis are carried out by external services requiring an execution time higher than a month. Accordingly, they can not be incorporated to the document in so brief interval.

Round 2

Reviewer 1 Report

Dear Editor

The manuscript well has been revised.

Reviewer 2 Report

N/A